# Exit Ripple Effects: Understanding the Disruption of Socialization Networks Following Employee Departures

## ABSTRACT

Amidst growing uncertainty and frequent restructurings, the impacts of employee exits are becoming a central concern for organizations. This study delves into the effects on socialization patterns between the remaining coworkers connected to departing employees. Using rich communication data from a large holding company, we track the longitudinal evolution of network metrics within communication subgroups of neighbors associated with exiting employees, contrasting these with the networks of neighbors of employees who stayed. We additionally compare these effects across two periods with varying degrees of organizational stress. We find evidence for a breakdown of communication of the socialization patterns of the remaining employees. These neighbors become more disconnected, less talkative, and their communication network more inefficient. The size of this reaction seems to be moderated by both external (periods of high organizational stress) and internal factors (characteristics of the departing employee). At the group level, periods of high stress correspond to greater communication breakdown. At the individual level, however, we find patterns suggesting individuals end up better positioned in their networks after a departure. This research provides critical insights into managing workforce changes and preserving communication dynamics in the face of employee exits.

## CCS CONCEPTS

• **Applied computing**; • **Information systems** → *Data mining*;

## KEYWORDS

Organizations, Communication Networks, Social Networks, Socialization, Employee Departures

**ACM Reference Format:**
Anonymous Author(s). 2023. Exit Ripple Effects: Understanding the Disruption of Socialization Networks Following Employee Departures. In *Proceedings of the ACM Web Conference 2024 (WWW '24), May 13–17, 2024, Singapore, Singapore.* ACM, New York, NY, USA, 13 pages. https://doi.org/10.1145/3543507.3583400

## 1 INTRODUCTION

Organizations are dynamic entities where personnel changes are an inherent feature. As employees depart, whether through voluntary resignation or enforced layoffs, there are undeniable consequences

on the structural and functional aspects of the organization. Prior literature indicates that these departures may impact employee morale, knowledge transfer, productivity, and other organizational outcomes [1, 32, 38, 46].

Quantifying the influence of an individual's departure on an organization has been challenging, often prompting researchers to depend on qualitative assessments [31, 32], narrowing to key employees [11] or to relating departure turnover rates to macro impacts on group and company performance [1, 38, 46]. In this study, we focus on the impact on socialization, specifically the network interactions among the remaining employees. For example, consider a scenario where Alice, Bob, and Charlie interact regularly at work. If Alice exits the company, how does this change the communication dynamics between Bob and Charlie? While our approach does not directly measure tangible outcomes like productivity or revenue, which are difficult to attribute to a single departure, it provides a clear and quantifiable way to understand the implications of personnel changes on interaction patterns and networks. Such interaction networks have proved indispensable for multiple context within companies such as development of organizational advantage [3, 37], collaborative task development [19, 33], and for the well-being of employees [39].

Furthermore, the uniqueness of our dataset allows us to scrutinize how departures influence socialization in the context of external factors and attributes of the departing employee. Notably, our data spans two distinct periods: one where the firm experienced stress and ambiguity, and another where it operated under more typical conditions. Drawing on findings from previous research which outlines how intra-organizational networks change with stress and ambiguity [17, 43], we analyze the interaction effects between the external stress level of the firm and individual departures. We find, for instance, that departures during periods of high stress is associated with less communicative groups in the company but it is also associated with patterns beneficial for individual employees.

In addition to external factors, we investigate heterogeneity of departing employee attributes. Gender, for instance, can influence communication styles and collaborative tendencies [12, 24], while a communication attribute such as closure within ones social network correlates with organizational knowledge transfer and influence [14, 42]. As such, we are particularly interested in exploring how these factors and others affect socialization dynamics post-departure.

Our research questions are thus: **RQ1** What is the effect of an employee's departure on the socialization dynamics of their prior contacts? **RQ2** How is this effect different during organizational high-stress periods? and **RQ3** How are attributes of the departing employee, such as their volume of communication or seniority, related to the response of the socialization set?

To frame our analysis, we draw on past research on social capital operationalized through networks [9, 39]. We employ a large-scale dataset of internal communications from a major company which

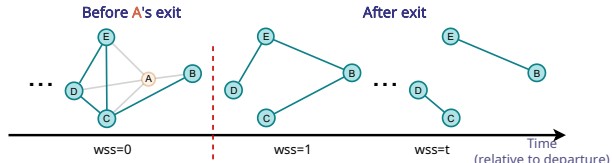

**Figure 1: We track the evolution of the interactions of neighbors of a departing employee. Figure exemplifies how the group changes its interactions after A's departure and ends up disconnected in two silos some time after. Here we only show the group perspective for simplicity.**

covers communications among 100K employees. From this data, we track changes in socialization sets of departing employees on a weekly basis, measuring various attributes associated with the group and individual dynamics over a period before and after departures take place. Following a model-based matched comparison approach, we assess relationships between departures and changes in socialization set attributes relative to a control group.

Our main contributions are as follows: **(1)** First large scale analysis of the dynamics of networks after node removal in an organizational context that also considers the effect of high stress organizational environments.**(2)** Our results suggests a breakdown of socialization among remaining members after a departure. The breakdown is marked by less connectivity, volume, cohesiveness, and efficiency when looking both at communications both between employees within the socialization set (group perspective) as well as interactions of each of these members within the larger organization (individual perspective). **(3)** We connect how the response to departures differs under periods of high stress. Notably, departures during these periods could be more detrimental to groups in the company but beneficial for individual employees. **(4)** Our research provides valuable insights for the field of organizational research by underscoring the ripple effects of a departure. We also connect the changes in the network structure after departures to observed consequences at the macro level, such as morale impact and group performance.

## 2 THEORETICAL FRAMEWORK AND RELATED WORK

Here we build a conceptual framework for our research informed by organizational literature about departures, socialization and social capital, and node removal in complex networks.

*Understanding Socialization Through Social Capital and Network Structures.* In our analytical framework, we use interactions in the communication data as a proxy for employee socialization. Socialization is crucial to onboarding and it is instrumental for employee effectiveness [21?, 22]. We argue that when people socialize within companies, they're building and using this social capital. One common lens to operationalize social capital is through networks [9, 39]. This is the lens we use in the present study.

We construct communication networks from our data. In these networks, we focus on the communication within what we define as the **"socialization set"** of departing employees. A person in this set is an organization member the departing employee would

interact with in normal day-to-day work circumstances. The interaction might be regular, as teammates or supervisors, or it could even be more sporadic. The inclusion of these sporadic ties is informed by the concept of "weak ties" [28] which are instrumental in information flow in organizations [10, 26]

*Dual perspectives on socialization and interactions.* The existing literature doesn't clearly connect employee socialization nor the mediation effect of a network node's removal to organizational departures. Nevertheless, it hints at possible reactions at different levels in the organization: At the broad company level, negative effects include decreases in productivity and a negative effect on the morale of remaining employees after key employee departures [1, 38, 46]. The effect on morale is also exacerbated by phenomena such as collective turnover and periods of high stress [8, 36]. However, there are caveats. Some research finds positive effects in performance when figures such as top managers leave [16, 29]. We contrast these departure 'group effects' with effects on individual employees. In some scenarios, departures can induce increased commitment and satisfaction from the remaining collaborators [32], or lead to positive outcomes such as increased job mobility [2, 40]. Some studies reason that the remaining employee having more leverage with the employer in order to cover for the missing position. This research also reveals that higher level collaborator exit correlates with more benefit for those remaining [2].

We note that departure might affect differently groups or individual employees. We also note how that the effects perceived are nontrivial and seem dependent on who leaves. Previous research however leaves out of analysis of the interaction effects between the remaining employees after a departure. In our study, we connect these ideas to the socialization interactions.

We bridge the previous literature with our research by adopting two central perspectives of interaction given the socialization set. The **"group perspective"** sees the socialization set of a departing employee ($e$) as a cohesive unit, focusing on the communication solely among its members. These interactions might be crucial even when they cross team boundaries as they are valuable for collaboration and socialization [22, 25]. Complementing this, the **"individual perspective"** considers each socialization set member's interactions beyond the set's boundaries, focusing on their broader network within the organization. Recognizing each person's diverse connections allows us to understand individual reactions and adaptations to an employee's departure.

*Network measures of social capital and correlation to networks.* We use the literature on social capital through networks perspective which explores the relationship between network structures and organizational out comes as well as the literature on node removal in networks to define measures in thew networks we are interested and to connect our results with previous literature.

Key concepts in social capital via networks are network closure and structural holes. We implement measures relating to these structures. Closure has been positively associated with team performance [4, 25] and with individual sense of cohesion and well-defined expectations [26, 39]. However it can also instigate a sense of coercion and self-segregation [10, 39]. Structural holes are associated with information and unconnected network spaces, offer

bridging opportunities. These network gaps can yield strategic benefits for organizations and individuals [9]. Employees acting as these bridges often achieve better career outcomes [9, 26] and allow for accessing disparate sources of information [9, 42]. Additionally, structural holes suggest increased flexibility within a network [26]. Closure and structural holes effects coexists in organizational networks contingent on connection types and connection content [25, 26, 39]

For the groups we also draw from literature on node removal on complex networks. Here characteristics that are important to understand in a group are efficiency, referring to information flow, redundancy [7, 34], and the size of the largest component [5, 6, 20, 30, 50] - are often considered to evaluate the impact of node removal. We adopt these measures in our analysis since we are interested in the effects on the group networks.

*Shocks to networks, turbulent environments.* Our review also extends to the literature detailing shocks and high-stress environments for networks. We use these insights to form hypotheses and interpretation of departure effects on high-stress times.

Researchers identify a phenomenon referred to as 'turtling up' when networks are subjected to abrupt shocks. The network restricts ties with out-group members, instead intensifying the cohesion of within-group ties [43]. In the face of changing environments, for instance, the shift to remote work, networks tend to become more compartmentalized [51]. When observing the evolution of within-group dynamics, compelling changes in tie composition among employees become apparent during periods of heightened uncertainty [18, 27, 48]. Researchers have broken down ties into formal, semi-formal, and informal networks and found that structural shifts often involve a decline in formal network ties, counterbalanced by an increase in semiformal and informal ties [18, 48].

## 2.1 Hypotheses

*Effect of an employee departure on their socialization set interactions.* From a group perspective, after a departure morale and efficiency could drop significantly [1, 38, 46], which for the group perspective network suggests decline in communication and efficiency [7, 34]. Furthermore, if we think that the departing employee might mediate the stability of a triad between coworkers, this connection might be compromised due to a node's removal [28], creating the potential for disconnection of the group perspective and formation of subgroups. On an individual perspective, reactions can vary. Some individuals might show increased commitment, fostering unity within their immediate circles [32]. Alternatively, others might display individual advantage [2, 40]. We can associate this to employees increase their communications, reduce their clustering and increase makers of brokerage within the company [9, 9, 42].

HYPOTHESIS (H1.1). *From a group perspective, the interactions of the socialization will break apart, marked by a decrease in communications and the number of connections within the group, as well as by a decrease in cohesiveness and closeness of the group.*

HYPOTHESIS (H1.2A). *From an individual perspective, on average, members of the socialization set will display a increased commitment effect marked by increased communication volume and increase clustering.*

HYPOTHESIS (H1.2B). *Alternatively, from an individual perspective, on average, members of the socialization set will display a individual advantage effect, where they increase their connections and volume and increase structural diversity.*

*Departures under high-stress environment.* According to literature, during high-stress periods, groups tend to isolate, forming disconnected components[51]. This implies a departure could worsen communication disruption from the group perspective, exacerbated by the stress-induced tendency towards disconnection.

HYPOTHESIS (H2.1). *During a high-stress period, at the group level we will observe increased group breakdown effect sizes*

The literature about individual employee adaptation in uncertain periods suggests individuals preserve and strengthen informal ties [18, 48]. Thus, following a departure, we anticipate that the resulting uncertainty escalated by both the environment and the employee's exit prompts individuals to maintain or seek diverse connections. This dynamic could also diminish clustering of connections, enriching diversity, and fostering increased network brokerage. However, alternatively, the effect of stress and hit to morale seen in turbulent times [36] might be significant enough that the individuals turtle-up showing increased clustering, and less communication [43]. This leads us to propose two competing hypotheses.

HYPOTHESIS (H2.2A). *During a high stress period, at the individual level, we will observed increased brokerage patterns.*

HYPOTHESIS (H2.2B). *During a high stress period, at the individual level, we will observed increased isolation patterns.*

## 3 DATA AND METHODS

We now proceed with detailing the data sources and methodology in order to test the hypotheses established in the previous section.

## 3.1 Data Context

Our research primarily investigates socialization patterns within a large company by using its internal communications data. This data also reflects a situation where the company transitioned to a high-stress period. The reason for this was significant regulatory changes that occurred in the early months of 2021. The country's government targeted the company's sector with regulations that, if adopted, could potentially make a significant part of their operations illegal. The threat and later imposition of these regulations created a sense of uncertainty within the company, resulting in significant workforce upheaval and attrition. Such effects may have influenced the company's internal communication practices during this period.

A vast portion of our data comes from the company's dominant communication network, which is an instant messaging tool akin to Slack. This data spans the period from January 2021 and represents 5M weekly interactions circulated among an estimated 120,000 employees. To ensure that our data only covers regular workday interactions, we excluded periods like holidays which could predictably influence communication patterns and introduce bias.

## 3.2 Networks

*Constructing the Weekly Communication Network.* We build weekly graphs, denoted as $\mathcal{G}^w = (\mathcal{V}^w, \mathcal{E}^w, \mathcal{W}^w)$, where $w$ denotes the week. This graph has nodes representing the set of all employees, $\mathcal{V}_w$, that have communicated within that week. Correspondingly, the edges in $\mathcal{G}^w$, denoted as $\mathcal{E}^w$ represent interactions between employees and have weigths $w_{ij} \in \mathcal{W}$ [15, 51, 52]. Every edge weight $w_{ij}$ is constructed with the aggregate volume of communications between $a$ and $b$ during that week. In our networks, we include both pairwise interactions (direct messages) as well as group interactions. The latter accounts for how groups maintain socialization and interaction in non-dyadic channels [51]. Specifics on weight calculations can be found in the appendix.

*Constructing Socialization Sets.* We denote the socialization set of an employee $e \in \mathcal{V}$ as $SS_e \subset \mathcal{V} = \bigcup_{w=t_e^*-10}^{t_e^*-6} \Gamma_{\mathcal{G}^w}(e)$, with $t^*$ marking the calendar week of departure of $e$. In addition, since we want to include interactions under 'normal' day-to-day work circumstances, we exclude interactions that might be part of the offboarding and handoff process that precedes a departure, thus we include only members that interact during weeks $[t^* - 10, t^* - 6]$.

*Estimating Employee Departures.* We use the information data to identify 40K employee departures in the year 2021. In essence, we estimate a departure by looking at the last time an employee appears in the communications data. Further details are in the appendix.

*Representing group and individual perspectives.* Formally, within the weekly communication networks, we define:

**Group perspective** as $G_{grp}^{e,w} = \mathcal{G}^w[SS_e]$, which is the induced graph given the socialization set. This represents the interactions only between members of the socialization group.

**Individual perspective** as $\{G_{ind}^{e',w}\}_{e' \in SS_e}$ where $G_{ind}^{e',w} = \mathcal{G}^w[e' \cup \Gamma_{\mathcal{G}^w}(e')]$, the ego network for $e'$. Thus the individual perspective is the set of induced ego networks (with respect to the whole communications network $\mathcal{G}^w$) of the neighbors of $e$.

Note the individual perspective of a neighbor includes the departing ego $e$. This is to highlight the individual socialization 'perspective' aspect of each neighbor. For instance, the leaving ego might be a key link for the cohesion of the neighbor, or perhaps it could be a member that if gone, would allow the neighbor to move to a strategic placement in the network.

## 3.3 Measures of socialization

We extract relevant metrics from the aforementioned individual and group perspectives on a weekly basis; for an approximately 32-week period centered around an employee's departure (16 weeks before and after). This process provides us with time-series data, $f^{e,m}(t)$, where $m$ represents the calculated metric, $e$ the socialization set index tied to the departing employee, and $t$ the time relative to the employee's departure.

For the **group perspective**, we calculate the following:

**closeness** Indicates how well connected is the group and it is akin to a measure of efficiency in the group [7, 34]. It is calculated as the average inverse distance of all pairs of nodes in $G_{grp}^{e,w}(G, e)$.

**cohesion** Indicates how cohesive is the group in the form of triadic closures [28] and is measured by the average clustering coefficient of $G_{grp}^{e,w}$.

**components** Indicates how many disconnected silos of interaction are within the group. Calculated as the number of components in $G_{grp}^{e,w}$.

**largest component share** A measure of network robustness [6, 20, 30, 50]. Calculated as $|G_{grp,0}^{e,w}|/|G_{grp}^{e,w}|$ where $G_{grp,0}^{e,w}$ denotes the largest component of $G_{grp}^{e,w}$.

**connections** Represents the number of pairwise connections within the socialization set members. Defined as the number of edges in $G_{grp}^{e,w}$ normalized by $|G_{grp}^{e,w}|$.

**volume** Represents the aggregate volume of interactions. Calculated as the sum of weighed edges in $G_{grp}(G, e)$ normalized by $|G_{grp}^{e,w}|$.

**n active** The number of active socialization set members during that week. It allows us to see if other members of the socialization also leave. Defined as $|G_{grp}^{e,w}|$

For the **individual perspective**, we calculate metrics for each $G_{ind}^{e',w} \in \{G_{ind}^{e',w}\}_{e' \in SS_e}$, which we then average to get one aggregate estimate. We describe now each measure in terms of each $G_{ind}^{e',w}$:

**clustering** Represents the embeddedness of $e'$ in their own network [13, 26, 39]. Calculated as the local clustering of the node $e' \in SS_e$ in the network $ego(e, G)$.

**connections** Represents how connected the employee is. Calculated for a given $e' \in SS_e$ as $|G_{ind}^{e',w}|$.

**volume** Represents overall employee communication. We take the sum of the edges of $e'$ in $G_{ind}^{e',w}$.

**diversity** Is an indication of how many communities $e'$ bridges and it is related to structural holes [15, 49]. It is calculated as the number of components in the graph $G_{ind}^{e',w}/e'$, ie. the components of the ego network of $e'$ removing $e'$ and its edges.

## 3.4 Matching

The inherent nature of corporate dynamics, coupled with prevailing uncertainties, can result in diverse and dynamic socialization patterns within a company. It is paramount to account for these externalities when analyzing the effect of an employee's departure on their socialization set. We can overestimate or underestimate the effect of employee departure if we ignore global trends in the data. For this reason, we incorporate a matching design to generate a control group that serves as a comparison for socialization sets of non-departing employees.

For each socialization set, we find a set of $k$ matches that are similar in departing ego attributes and socialization set metrics. We generate an estimate of these attributes given an employee $e$ by averaging the metrics over a defined period of time $[t_e^* - 10, t_e^* - 6]$, which corresponds to the same period of time on which we defined the socialization set $SS_e$ relative to $e$'s departure. We then perform a filtering of the matches week by week of departure to balance

the groups. In addition, due to computational limitations, we perform the match on a proxy of the attributes of the socialization set. Further details about this procedure are in the Appendix.

## 3.5 Models

*Model definition.* To quantify the average change in the metrics of the socialization sets post the departure of their corresponding ego employees, we develop a model-based approach. For each metric denoted as $f^m$, we fit a model to encapsulate its dynamics. It models the time progression of the metrics with linear trends relative to the departure timing of the socialization set's ego $e$ [23]. Our model incorporates a linear splines basis which allows for modeling changes in value and trend of the metrics after departure [45], random effects to account for the variability between different socialization sets [41], and uses the matched socialization groups to perform before-after contrast comparison of response in metrics using a control group [44]. The model is expressed as follows:

$$f_{e,t}^m \sim A_e \times (t + hinge(t) + jump(t)) + controls_e + \eta_{e,t} \quad (1)$$

Where we have functions that define the time basis: $hinge(t) = t * \mathbb{1}(t > 0)$ denotes a change in slope and $jump(t) = \mathbb{1}(t > 0)$ which denotes a discontinuity in value at time $t$. Here, $e$ denotes each socialization set $SS_e$, which is itself defined by the departing ego $e$, $t$ corresponds to the relative time to departure, $A_e$ is an indicator with a value of 1 if the ego employee of the socialization set departs after $t = 0$, serving as a marker for our treatment group. Then, $\eta_{e,t}$ encodes a random effect by departing ego/socialization set which accounts for baseline metric variations among distinct employee departure groups. Finally, by incorporating an interaction between the treatment indicator and time basis, we can compare the treatment and control groups' relative discontinuity and slope differences using marginal estimates which we describe in the next section.

For model fitting, we transform target variables to allow for comparison between different metrics and adjust for heavy-tailed variables. Further details can be found in the appendix.

*Model estimates.* After model fitting, we quantitatively measure the response of socialization sets using two estimates extracted from our fitted metrics models:

**Value DiD**  Denoted as $DiD_{val}(\hat{f})$, contrasts how the metric changes from the pre-departure to departure period compared to the control group. We calculate it as the difference (between groups) of the difference in each group comparing before and after values of the estimates of a metric. Thus for example a positive value of $x$ in the metric $f$ means that compared to the control group, the socialization sets of departing employees change $x$ more relative to the control group change.

**Slope DiD**  Denoted as $DiD_{slp}(\hat{f})$, captures the metric slope difference between pre and post-intervention for each group estimate. These estimates reveal relative metric changes and are indicative of whether they increase or decrease over time. For Illustration, a positive value of this estimate indicates that the change in trend for the treatment group was larger when compared to the control.

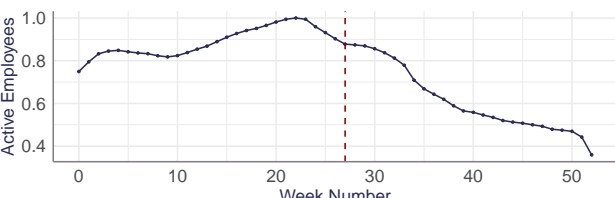

**Figure 2: Weekly active employees in the communication data. The red line indicates the week on which the high-stress period starts. Counts for 2021, normalized by the maximum value.**

For further details including numeric expressions for the calculations of these estimates, please consult the appendix.

These two estimates allow us to identify both a change in the value of a metric after departure and a trend that indicates if this change would be persistent over time or rather would go to the status quo. We can define quadrants that indicate possible situations contingent on the signs of Value DiD and Slope DiD. We display these quadrants alongside main results in Figure 3.

*Assessing the effect of uncertainty.* In our study of changes in the network structure of socialization sets amidst heightened uncertainty, we exploit the unique aspect of our data where the government enacted a policy that initiated a period of high uncertainty accompanied by reorganization. This lets us separate the data into two periods as seen by the dotted red line in Figure 2. We refer to the period post-implementation of the ban as the high-stress period. We split our data into two sets, representing employee exits in each period, with July's first week as the dividing point. A one-month buffer was applied before and after this date, removing any employee exits within this period from the data. Consequently, we have 1M observations for 80K treated and control socialization sets for each period.

We apply the same model described in the previous section to each period separately. Using these fitted models, we calculate diff-in-diff estimates for both value and slope, then compare them between the two periods, illuminating changes in network structure dynamics.

*Model for heterogeneous effects.* Our third research question revolves around potential variations in the socialization sets after departures that might be associated with different characteristics of the departing ego individuals. We calculate attributes from additional data sources provided by the company. Specifically, we examine the following ego characteristics: **leadership status, seniority, and gender**. The communication-related attributes we calculate are: **ego's volume of communication, number of connections, clustering, and structural diversity**. These metrics are essentially the 'individual perspective' of the departing ego. For additional details on the definition of these variables please consult the appendix.

To model the differences in each of these ego attributes, we employ a slight variation of the previous models. In this case, instead of contrasting treated and control socialization sets, we contrast levels of the attributes. For example, we contrast how the response to a departure differs when comparing a highly clustered departing employee to a low-clustered employee. Or a senior departing

employee compared to a non-senior. Details about the model and estimates definitions can be found in the appendix.

## 4 FINDINGS

### 4.1 RQ1, Effect of ego's departure on the socialization set

Addressing RQ1, we calculated the model-based value and slope DiD estimates as outlined in Section Section 3.5. Figure 3 presents these estimates on two axes: one for value DiD and another for slope DiD. We report these coefficients in terms of the standard deviation of the metric across all observations (DM) and of the rate of change of this unit per week (DM/w) for value DiD and slope DiD respectively. All values reported are significant (p<0.01).

We observe significant coefficients for all the metrics, supporting our hypotheses that socialization sets experience an impact on ego departures when compared to control. We now delve into the nature of this effect from both group and individual perspectives.

From a group perspective, **we find support for hypotheses H1.1**, we observe an increase in DiD values for closeness, cohesion, connections, and communications volume. Cohesion and closeness were the metrics most impacted within group socialization with a comparative decline of -0.33 DM and -0.42 DM respectively. Following the departure of the ego employee, there is also a significant decrease in the number of active members within the socialization set of -0.646 DM. It's important to note that the decrease in communication volume and connections is not solely due to the reduction in active members within the socialization set, as these variables were normalized by node count, and the model controls for this factor. This indicates a compounded decrease in volume and connections, even with a reduced number of active nodes.

We also examined the Diff in Diff of slopes to understand temporal dynamics. For the group perspective, connections (0.0113 DM/w) and share of the largest component (0.0131 DM/w) display a positive slope, in contrast to their negative value DiDs. This suggests the potential convergence of certain metric differences over time between the treatment and control groups. Contrasting this, metrics such as group closeness (-0.0257 DM/w), cohesion (-0.0139 DM/w), and number of components (-0.0123 DM/w) show a negative slope DiD, indicating an increasing difference between the treatment and control groups over time. Overall, this implies that treated socialization sets might continue to experience decreasing cohesion and communication volumes.

On the individual perspective, we note a decrease in value DiD for individual connections (-0.0871 DM), volume (-0.1127 DM), and diversity (-0.0668 DM). Compared to other metrics, however, the decrease in diversity is relatively minor. Additionally, there is an uptick in the clustering of neighbors (0.1113 DM) within each socialization set member's personal ego network. Unlike group metrics where the ego was not included pre-departure, immediate changes are expected in individual metrics, as they factor in the departing ego. Nevertheless, we shed light on these changes' effect sizes, pinpointing the most impacted metrics. In this case, the most impact occurs in individual clustering and volume.

Looking at the slope DiD for changes over time, we find that individual employees seem to rebound to the status quo relatively quickly in terms of volume and communication levels as evidenced

by the higher value of slope DiD and the position in quadrant II. However, increased clustering within their communication networks (Quadrant I) and stagnant diversity (minor 0.0032 DM/w estimate) become apparent. These observations may imply potential challenges for individuals as brokers of information.

### 4.2 RQ2, Response under heightened uncertainty

In our investigation of RQ2, we delve into the responses of socialization sets for departing employees during two distinct periods. Figure 4 displays overlapping value and slope DiD estimates for both periods, with a brighter orange for the period of increased stress. Our focus is to discern relative differences in these estimates.

Results in Figure 4 here suggest more drastic effect sizes in socialization patterns from the group perspective during periods of heightened uncertainty, evidenced by larger effect sizes in the metrics estimates of roughly 3-4 times during high stress. **This supports hypotheses H2.1**. Group estimates typically align in direction (same sign) in the two periods, but distinct variations in magnitude emerge. Predominantly, group closeness and cohesiveness display significant negative value DiD estimates under high stress of (-0.3756 DM) and (-0.3636 DM) respectively. These values are roughly 3x-4x larger than the values during the low-stress period. However, slope DiD estimates remain consistent across the periods. With group components, a noteworthy reversal appears, whereby the value DiD progresses from non-significant during a lesser stress period (0.0097 DM) to slightly negative under higher stress(-0.0646 DM3). Group connections also show a likely reduction in magnitude from -0.0423 DM to non-significant during high stress. Group communication volume has a larger negative effect size under higher stress, from -0.0309 DM to -0.0969 DM but rebounds swiftly as suggested by the positive slope DiD (0.0128 DM/w).

Moving to the individual perspective during high stress in Figure 5, we find that, during high stress, the individual perspective shows decreased individual clustering and increased communication and diversity. These findings support Hypotheses 2.1. First, the individual clustering value DiD is negative pre-stress (-0.0393 DM), with a non-significant effect during high-stress. In addition, slope DiD significantly increases under high stress (0.0158 DM), suggesting a gradual divergence. Individual connections and volume show minor negative estimates under less stress but a large positive value DiD for high stress. These estimates are around 4 times larger than the pre-stress estimates. Diversity also presents a large reversal from negative (-0.0902 DM) to positive (0.0668 DM).

### 4.3 RQ3, Heterogeneous effects

In examining RQ3, we analyze nuances of impact within socialization groups, considering departing ego attributes. Figure 6 visually depicts DiD estimates from the ego attribute interaction model, with color intensity signifying the estimated value (darker hues represent higher values).

For example, observe the cell corresponding to the DiD value for seniority contrast (column) and group cohesion metric (row). Here, DiD estimates enable the comparison of leaders against non-leaders (DiD calculated as $egoleader - egononleader$). A positive coefficient, here 0.1 DM, indicates that if the departing employee is a leader,

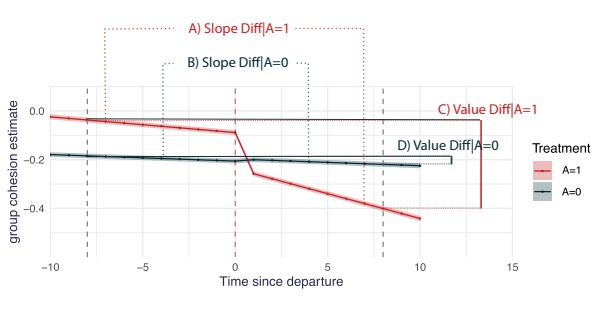

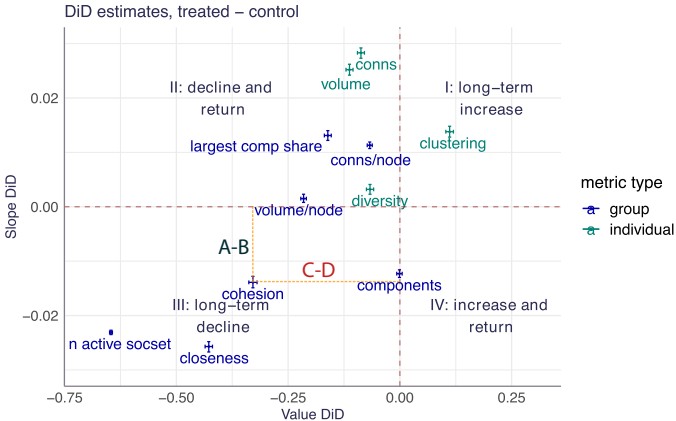

**Figure 3: L) Sample of marginal estimates for group cohesion across time. Also shows pictorially what would correspond to value and slope differences per group. These are aggregated into the DiD as also shown on the right. R) Quadrants with model-based DiD estimates (Section 3.5) of** *treated − control* **socialization sets according to our model Eq. (1). Most estimates are located in the quadrant of negative value and trend, indicating communication breakdown at both the group and the individual perspectives.**

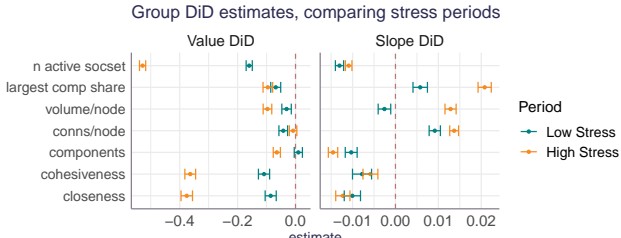

**Figure 4: Comparison of estimates between the periods of low vs. high stress. Larger effect sizes for the group perspective indicate larger breakdown-related effect sizes**

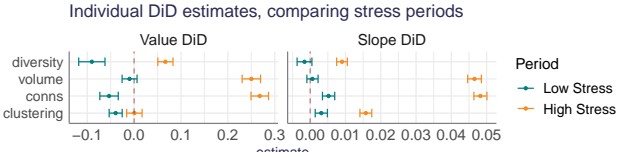

**Figure 5: Comparison of estimates between the periods of low vs. high stress. Individual perspective shows a reversal where individuals are more diverse and have more connections and volume.**

the socialization set group cohesiveness is larger by 0.2 standard deviations of the metric (DM) than if the departing employee was a non-leader. In other words, compared to non-leaders, departing leaders leave socialization sets that end up more cohesive.

We find that ego features associated with the most pronounced differences are the ego's clustering, diversity, leadership status, and leader seniority. A few highlights: Higher ego clustering is associated with increased group communication breakdown as evidenced by the negative values across group metrics. Departing employees with higher diversity are associated with further decline in communication metrics, but interestingly, the number of components

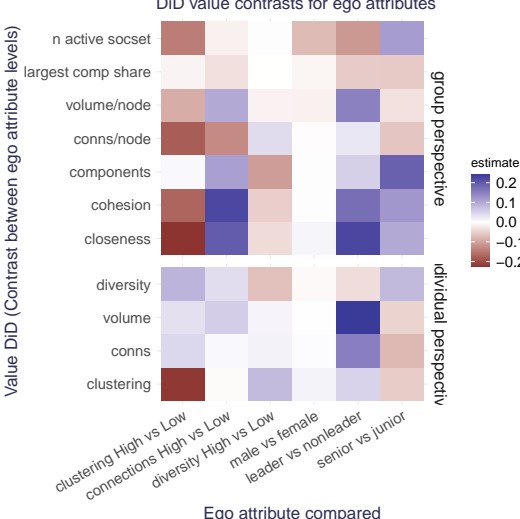

**Figure 6: Value DiD estimates from the ego attributes interaction model. Cell** $(i, j)$ **is the Value DiD comparing the levels of the ego attribute** $j$ **for metric** $i$**. For example, i=group cohesion, j=ego leader vs. non-leader has a positive value of around 0.1. Thus socialization sets where the departure is a leader have increased cohesion compared to a non-leader departure).**

decreases and the largest component size increases, meaning that the group becomes more connected, but communicates less overall. Regarding gender, our estimates find no support for the ego's gender being associated with significant differential effects. That is, for this model, the change in slope and value is the same for socialization sets where the departing ego was male or female. Finally, the departure of seniors and leaders has a similar effect where

the group breaks up into communicative silos (as evidenced by increased components but also increased connections and volume of communications).

## 5 DISCUSSION

*Breakdown of socialization after employees departure.* We find a significant breakdown in the socialization set interactions after an employee's departure, supporting hypotheses **H1.1** and **H1.2**. This disruption is characterized by a decrease in communication volume, connectivity, cohesion, and efficiency in the group perspective, leading to fragmentation into isolated silos. From the individual perspective, members tend to become less communicative, establish fewer connections, and exhibit increased clustering and lower diversity. These observed breakdown effects persist over time, despite a potential rebound to the status quo in some of the metrics. These results indicate that there is more to resignation than just losing the employee, given the loss of social capital enabled by their connections. This is particularly relevant in high collaboration contexts, where both intra-team and inter-team communication is critical [19, 33]. In these contexts, our finding of reduced efficiency in communications of groups could explain the findings of other studies that observe a decline in performance in groups after resignation [1, 46]. Our findings on the reduced connectivity in the individual perspective may also have implications for the literature on employee morale [8, 36] since interactions among colleagues also entail emotional support [25 **?** ].

*Potential mechanisms mediated by departing employee attributes.* Based on our results from the perspective of triadic closure [25, 28, 39], we can propose a potential mechanism behind the aforementioned breakdown. Suppose A, B, and C are connected by triadic closure. When A departs, the closure between B and C is weakened. Thus, it is more likely to observe B and C not communicating as frequently, or at all in the future. Extrapolating to a socialization set, the departing employee holds several 'closures' between its neighbors. In our analysis of heterogenous effects, we observe that compared to less clustered departing employees, when departing egos are more clustered (more closures), there is a stronger communication breakdown of the group. Similarly, we also note that for highly diverse departing individuals that bridge different groups, the group displays increased connections and a reduction of silos, but also lower cohesiveness and efficiency. In other words, the group attempts to reconnect the silos of information but has a less efficient structure. This suggests a mechanism of adaptation where the network attempts to recover their procedural connections and information flow [5, 9, 10].

*Increased stress exacerbates group breakdown, but also individual advantage.* In studying socialization dynamics of departures during high-stress periods, we found that from the group perspective, the effect sizes related to communication breakdown after the departure of an employee are larger. However, interestingly, from the individual perspective, we find that individuals start to communicate with more connections and show higher structural diversity. These patterns suggest that the departure of a connection leads to an individual's structural advantage. These findings are consistent with previous research where the higher communication and diversity are correlated with the individuals assuming an information brokerage position that is advantageous for them [10, 39, 47].

*Implications for broader network research.* Our research not only aligns with research on organizations but also informs, more generally, the impact of node removals in networks[5]. Our results highlight the complex, dynamic responses to node removals by connecting the ideas of triads, cohesion, and the network's adaptation to maintain operations. They also reveal how sometimes outcomes are detrimental for the group but beneficial for individuals. This study can potentially enhance our comprehension of node removals across various types of networks.

*Limitations.* The study has several limitations. First, we do not employ direct performance measures such as employee productivity. Instead, we rely on previous literature to describe potential consequences of particular network structures, such as associating increased individual structural diversity with increased advantage within the organization. Nonetheless, we note that understanding the effect on the networks provides insight into aspects that are usually harder to measure directly, such as the cohesion of a group. Second, we do not take into account a categorization of interactions, such as differentiating between formal and semiformal ties, or distinguishing team boundaries. These categorizations have been applied in prior research to offer a richer view of network structure effects [25, 39]. Although we recognize the benefit of this nuance, we were interested in a more general view of interactions as a first approach to our research questions. Finally, we remark that our approach does not establish causal relationships. We however use matching to give better contextualized estimates of the potential effects of departures.

## 6 CONCLUSION AND FUTURE WORK

Our empirical study sheds light on the effects of a departure in the socialization of peers. This exploration rested upon a networks perspective and utilized vast readily available internal messaging data. Post-departure, we find evidence of a significant disruption in the socialization patterns of the remaining employees. The size of this reaction seems to be moderated by both external factors, such as periods of high stress, as well as ego-centric factors, such as the level of communication or seniority.

Future studies could aim to establish the effect sizes on performance characteristics thought to be linked to social capital. There is also a clear opportunity to parse out the effect of resignations on different types of network ties, according to content. Following this, a greater understanding could be sought to determine if the effects we observe can be generalized to other types of networks.

Finally, when juxtaposed with organizational literature, we argue that the theoretical frameworks and mitigation practices could be enriched if they considered the ripple effects of a departure on the interactions of other team members. This implies that beyond replenishing capabilities through replacement, the interaction dynamics amongst remaining team members should be accorded significant consideration. Furthermore, our research findings provide salient empirical evidence for scholarship on the removal of nodes in complex networks that dynamically respond to such changes.

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

# A APPENDIX

## A.1 Networks construction

*Weights of interactions.* In a given week, any direct communication between two employees $i, j \in \mathcal{V}$ has a weight $w_{i,j} = 1$ and is aggregated over the week. In addition, we assign edge weights in group interactions in the following manner: If employees $i$ and $j$ are part of an interaction involving $k$ participants, $w_{i,j} = 1/k$. To give an example, in a 10-person meeting, a message weighs 0.1 to each recipient. One can think that in a group of $k$ people, a message is replicated $k$ times, but the importance is diluted in the $k(k-1)$ possible pairs. This leaves roughly of $k/(k(1-k))$ $1/k$ messages dispersed within the pairs.

*Pinpointing employee departures.* We identify an employee's departure when they cease to participate in IM communications, leveraging the prevalence of the IM channel as the defacto communication channel according to company representatives. We remark that for the period of 2021, this inference is robust since we have communications data until mid-2022. Thus, eliminating possible false positives due to vacations or similar phenomena.

## A.2 Matching details

For each employee $e$ that leaves the organization at a particular week, we find other $m$ employees $e'_m$ who at the time of $e$ leaving, were still present in the company. We also look for $e$ that are similar to the departing $e$. We establish similarity both on the basis of similar socialization sets and on similar communication characteristics of the departing employees themselves. The reasoning for the latter is to take into account possible variations of the behavior of the egos that might imply different responses after resignation. For example, one can envision a scenario where two employees, both with an equal number of network neighbors, exhibit different interaction intensities. One of the employees soon to exit might exhibit a more active communication behavior or demonstrate a higher degree of cohesion within their network.

*Matching features.* We have determined the following set of attributes that form our matching criteria for the ego communication patterns: The number of connections of the ego, clustering of the ego network, and a binary variable that denotes if the employee is a manager or not. These are in other words, the individual perspective metrics of the ego that indexes the socialization set. Then, to aggregate the estimates over time to give a more stable set of matching measures for the egos, we take the 4-week average of the measures over the freeze period of identification of contact neighbors, that is, the average of the metrics for the weeks that correspond to a time interval $[n_{buffer} - n_{freeze}, n_{freeze}]$. We end up with a dataset of averages of the aforementioned measure for each employee in the organization. This is the input for the matching algorithm

For the matching algorithm, we use k nearest neighbors (kNN) in Euclidean space. Upon collating the attribute matrix, we standardize the features ensuring they conform to a mean of 0 and a standard deviation of 1. Our approach identifies the top $k$ matches from the subset of non-departing. This process is repeated weekly for all departing employees of that week. We also make sure that the matched employee is not a direct neighbor of the target employee. That is, that at the time of comparison of initial states, there is not an edge of IM communication connecting the match and leaving employees. We do this to remove the correlation that direct neighbors have as they likely interact with each other and affect each other's communication patterns.

Given that in our dataset, a significant portion of employees departed, it happened that at selecting k matches for each leaving employee we ended up selecting the same matched egos $e$ multiple times. Note however that it could be that an employee $e$ was selected at two different weeks, which means that it corresponds to two different socialization sets. Still, repeated matches induce artificial autocorrelation, which is especially concerning if the matched egos get selected over weeks that are too close. To alleviate this issue we incorporate two different procedures for filtering matches. The first one includes a step where the top-k matches are selected randomly from a top-k' where $k < k$. In our case, we used $k = 3, k = 20$. This is in order to introduce more variance in the selected matches for each leaving ego $e$. The second one is a post-filtering where week by week we subset the population of matches by selecting a sample of the matches for that week such that the matches have not been used recently within a particular window of time $w_{recent} = 4weeks$. This last one directly helps reduce autocorrelation in observations for matches of the same ego that could be too close in time.

Then, with the matching calculated, to assess matching quality we inspect the distribution of distances to matches Figure 7, and compare the distributions for both variables used for matching ?? and the metrics of interest of the socialization sets among the resulted treated and control groups Figure 8. We perform this comparison through visual inspection of the distributions.

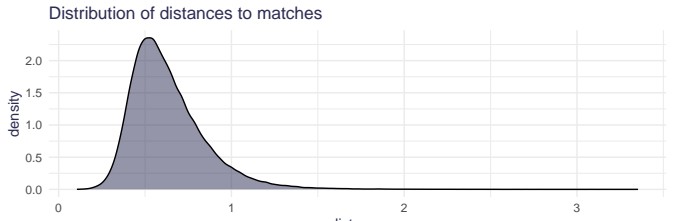

**Figure 7: Distribution of weekly distances between treated and matches. These distances are with respect to the space of the features used for matching.**

## A.3 Modeling Details

*Mathematical definitions of Value DiD and Slope DiD.* We now provide mathematical definitions and computation details for the model estimates.

First, we define $DiD_{val}(\hat{f})$ by

$$DiD_{val}(\hat{f}) = D(\hat{f}^m|A = 1) - D(\hat{f}^m|A = 0) \qquad (2)$$

Where $D$ corresponds to a difference between periods, within the same group

$$D(\hat{f}^m|A = a) = \hat{f}^m(t = t^+|A = a) - \hat{f}^m(t = t^-|A = a) \qquad (3)$$

Computation of this involves contrasting estimates between weeks $t^+ - t^-$. We select these as 8 weeks before and after departure within groups since $t^- = t^*_e - 8$ falls in the middle of the period of

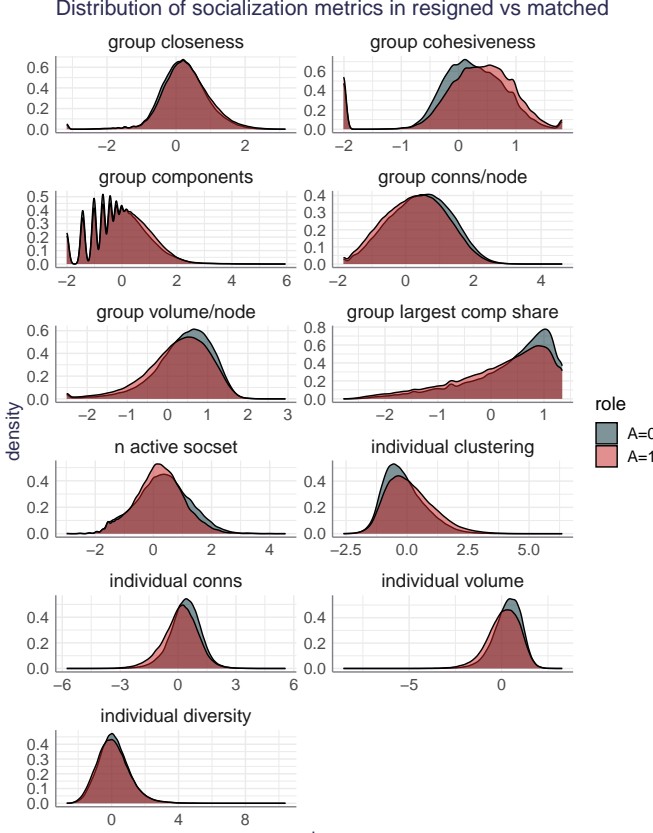

**Figure 8: Distribution of target metrics of the socialization set comparing treated and control socialization sets. The distribution plots only values before departure**

the definition of the socialization set. Then, we contrast the time differences between treatment and control socialization sets.

Second, the estimate for the slope DiD:

$$DiD_{slp}(\hat{f}^m) = \partial_t D(\hat{f}^m | A = 1) - \partial_t D(\hat{f}^m | A = 0) \quad (4)$$

captures the metric slope difference between pre and post-intervention for each group estimate. These estimates reveal relative metric changes and whether they increase or decrease over time. For instance, a negative $DiD_{val}$ and $DiD_{slp}$ for cohesion suggest cohesion decreases comparatively in the treated group, a decrease that worsens over time. We obtain these estimates with the R package *emmeans* [35].

*Target transformations.* To facilitate a nuanced interpretation via comparison of change across the different metrics and model measures, we employ two transformations to our target metric variables.

- A log transformation $\log(f) + 1$ to address heavy-tailed behavior in the metrics volume, connections, and neighborhood size.
- z-scoring on our target variables to standardize the data, enabling us to compare the magnitude of changes across diverse metrics.

These quantities become the dependent variables $f^m$ in our models. Leveraging this approach allows us to contextualize metric alterations using a scale defined by standard deviations of the population. This allows to estimate adimensional effect sizes and to compare effect sizes across different metrics.

*Heterogeneous effects attributes.* Here we outline the definitions of the attributes that we used in the analysis for RQ3

*leader* is a binary variable that signifies whether an ego assumes a leadership role. To determine this, we leverage data from the org chart during the months of July to August and assess whether the ego appears as a leader during this period.

*Senior* denotes whether a leader has more than five years of experience after college, corresponding to the 75th quantile of this variable in the data. This information is sourced from a company-provided dataset, which offers a snapshot of this data.

*gender*, in this dataset, is represented as a binary variable indicating whether an employee is registered as male or female within the company. Like seniority, this information is also derived from a single snapshot.

On the other hand, for communication attributes of the departing ego we use measures for the following: ego's volume of communication, number of connections, clustering and structural diversity. These measures are essentially the 'individual perspective' of the departing ego, as defined in Section 3.3.

*Heterogeneous effects model.* For analyzing heterogeneous effects, first, we only fit these models with data pertaining to socialization sets with indexing egos that have left the company. In other words, we restrict our analysis to the treated group. The model takes the following form

$$f_{e,t}^m \sim X_e \times (t + hinge(t) + jump(t)) + controls_e + \eta_{e,t} \quad (5)$$

Where $X_e$ denotes the set of attributes of the departing employee $e$. The model includes interactions between each ego attribute and the time basis. For analysis, we generate marginalized difference-in-differences (diff-in-diff) estimates between levels of each ego attribute. This lets us contrast estimates between levels of the attributes For instance, when examining gender, we compute the following estimate:

$$DiD(\hat{f}^m) = D(\hat{f}^m | gender = Male) - D(\hat{f}^m | gender = Female) \quad (6)$$

where,

$$D(\hat{f}^m | A = a) = \hat{f}^m(t = 8 | A = a) - \hat{f}^m(t = -8 | A = a) \quad (7)$$

*Level comparison estimates for heterogeneous effects.* For ego attributes that are included in the model as continuous variables, such as ego volume of communication, we generate estimates to compare the 1st and 3rd quartiles of the distribution. In essence, we investigate how lower-volume departing egos compare to higher-volume egos, as defined by the distribution. For the case of ego volume, the Diff-in-Diff estimate is computed as:

$$DiD(f_m) = D(f_m | vol = Q3(vol)) - D(f_m | vol = Q1(vol)) \quad (8)$$

This analysis provides insights into the varying response contingent on different ego attributes on socialization network metrics during the defined time frame surrounding ego departures.

## A.4 Results Appendix

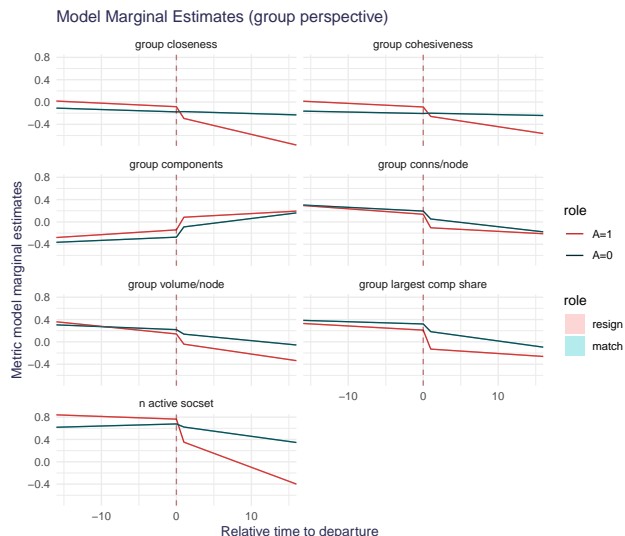

Figure 9: Plots with model marginal estimates of the metrics including the effect over time. Using the models for RQ1.

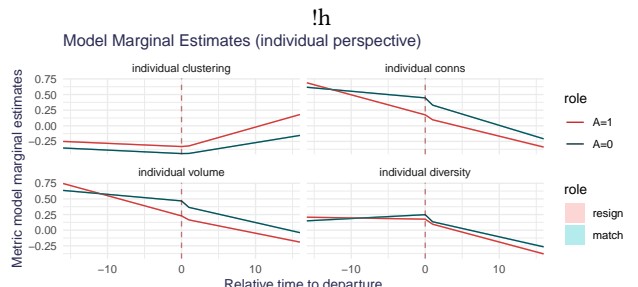

Figure 10: Plots with model marginal estimates of the metrics including the effect over time. Using the models for RQ1.

