# OpenReview forum: "Exit Ripple Effects: Understanding the Disruption of Socialization Networks Following Employee Departures"
_ACM.org/TheWebConf/2024/Conference — TheWebConf24_

### Official Review · Reviewer_nqZS · 2023-11-24

**Novelty:** 5
**Technical Quality:** 4

**Review:**

The paper aims to investigate the effects of employee departures on socialization patterns within organizations. It analyzes communication data from a large holding company and examines the changes in network metrics for coworkers connected to departing employees. The study also explores the impact of organizational stress on these effects.

The paper is also well-structured and well-written.

My main concern is the generalizability of the analysis, as only one dataset has been examined. It would be beneficial to investigate whether similar patterns exist in other datasets.

Additionally, could the authors provide a more comprehensive definition of "high-stress" and offer more examples to enhance understanding?

Furthermore, the model metrics are estimated based on the control group. How can we ensure that the implemented changes did not impact the metrics in the control group?

**Questions:**

please refer to above questions.

**Reviewer Confidence:**

2: The reviewer is willing to defend the evaluation, but it is likely that the reviewer did not understand parts of the paper

**Scope:**

3: The work is somewhat relevant to the Web and to the track, and is of narrow interest to a sub-community

---

### Official Review · Reviewer_AHdo · 2023-11-29

**Novelty:** 6
**Technical Quality:** 4

**Review:**

The paper examines the effects of employee exits on the socialization patterns in a large company. It utilizes communication data from Slack to explore the longitudinal evolution of network metrics, particularly focusing on how these dynamics shift in the wake of employee departures and during periods of varying organizational stress. The research finds that employee exits lead to a breakdown in communication patterns (as expected), with remaining employees becoming more disconnected and their communication networks becoming less efficient. This work is crucial as it delves into a relatively underexplored area of how personnel changes impact the intricate web of workplace interactions and offers valuable insights for managing workforce changes.


1. Strengths:
    * The study is probably the first of its kind. Even though some of the insights are trivial, some of them are really helpful and could potentially be helpful in studying organizational behaviour.
    * The study’s use of comprehensive communication data to track changes over time is commendable. This data-driven approach allows for a nuanced analysis of socialization dynamics and provides a solid empirical foundation for the study.
    * The comparison of employee exit effects across different stress periods within the organization is particularly insightful. It provides a deeper understanding of how external factors influence internal communication dynamics

Weaknesses:
* The study focuses on a single company. This specificity may limit the generalizability of the findings to other organizations with different cultures, sizes, or sectors.
* While the use of communication data is a strength, the study might benefit from a more qualitative or mixed-methods approach to capture the nuanced human elements of socialization that may not be fully represented in quantitative data.


I would strongly encourage the authors to release the dataset (anonymised appropriately) if possible. This will help spur research building on such valuable data.

Translating the findings into actionable strategies for organizations could be a valuable next step. This could include recommendations for maintaining effective communication and preserving network efficiency during periods of high turnover or stress.

The study hints at individual-level changes in the wake of employee departures. Delving deeper into these individual impacts could provide a more comprehensive understanding of how departures affect employees on a personal level.

**Questions:**

I have no comments/clarfications.

**Reviewer Confidence:**

3: The reviewer is confident but not certain that the evaluation is correct

**Scope:**

4: The work is relevant to the Web and to the track, and is of broad interest to the community

---

### Official Review · Reviewer_96ph · 2023-11-29

**Novelty:** 5
**Technical Quality:** 5

**Review:**

Summary: The paper attempts to study the effect of departing employees on the organization by focusing on the amount / nature of communication between employees as a proxy. The authors ask several interesting research questions - what is the effect of departing employees on their immediate neighbors (in the communication graph)? Does a departure help or hurt the group dynamics? How much does seniority / individual characteristics affect these results? What is the effect of high stress environments on these analyses?

The authors conduct a thorough empirical study using data from internal messaging service of a large company. Given the challenging nature of the work, the authors are forced to use a number of proxy measures to determine important quantities such as the departure time of an employee, the productivity of the remaining employees, impact of the employee on group dynamics, etc. The paper is well-written and uses sound methodology / justifications and a thorough discussion of limitations.

**Questions:**

Q. Is TheWebConf an appropriate venue for this paper? The paper feels very tangentially related to the web.

Q. How robust are the results to different choices of network construction? For example, assigning a weight of 1/k to a group message with k people is pretty ad-hoc. Even in medium-to-large groups, often a large number of conversations are back-and-forth between a small number of people; distributing the weight of such communication equally with the group is not necessarily the right approach.

Q. Have you considered a similar analysis of new employees joining the organization?

**Reviewer Confidence:**

3: The reviewer is confident but not certain that the evaluation is correct

**Scope:**

3: The work is somewhat relevant to the Web and to the track, and is of narrow interest to a sub-community

---

### Official Review · Reviewer_VNpY · 2023-12-01

**Novelty:** 5
**Technical Quality:** 3

**Review:**

This paper seeks to investigate the effects of an employee's departure from the organisation on the communication dynamics of the departing employee's prior contacts in the organisation. The paper investigations these effects during high stress period, for instance, during layoffs. Also the paper looks at how these effects vary depending of attributes of an employee, e.g., seniority. For the investigation, the paper studies communication within an organisation with over 100K employees. The results show that a communication breakdown occurs after employees depart.


Strengths:
- Overall a good paper making a relevant and interesting contribution with broad applicability across domains.
- Very large scale study to study communication patterns: 120,000 employees and 5 million weekly interactions. This large scale helps generalisability.
- Extensive experiments to study various effects.

Weaknesses:
- No mention or discussion around obtaining ethics approval or consent from employees.
- Reproducibility of such a large-scale study is challenging. What all was collected and what all was used needs to be made explicit. How certain parameters were selected need justification. e.g., in 3.2, why [t-10, t-6] range?
- The experimental results need better statistical grounding.
- Some terminology usage is unclear. E.g., p-values are used in 4.1 and also there is mention of effect size for statistical significance. "effect size" is a statistical measure. It is unclear if "effect size" in the paper refers to statistical effect size or is it just a metric name with no relation to statistical effect size. Also, p-values are often criticised. Consider taking a look: McShane et al., "Abandon Statistical Significance." https://arxiv.org/abs/1709.07588

**Questions:**

- How was consent obtained from the employees? Were they aware that their communication patterns were being studied? How was consent obtained from the employees who departed the organisation?
- In 3.2, 3.3, 3.4: What is the rationale for selecting 32 week period around an employees departure? Why metrics are computed for period of time [t-10, t-6]? Were these numbers empirically computed based on certain analysis or observation, or were these arbitrary?
- Ref RQ1: For significance p<0.01 is considered. As multiple hypotheses are being considered together, was Bonferroni correction applied?
- Ref RQ2: Does effect size in 4.2 refer to statistical effect size? If so, which effect size measure is computed?

**Ethics Review Description:**

This paper presents a large-scale study of in-organisation communication in a company with 120,000 employees. The paper, however, does not provide any detail about obtaining consent from the participants (employees of the organisation).

**Ethics Review Flag:**

Yes

**Reviewer Confidence:**

2: The reviewer is willing to defend the evaluation, but it is likely that the reviewer did not understand parts of the paper

**Scope:**

4: The work is relevant to the Web and to the track, and is of broad interest to the community

---

### Official Review · Reviewer_RLxX · 2023-12-01

**Novelty:** 4
**Technical Quality:** 3

**Review:**

The paper studies the impact of socialization and network interactions between the remaining co-workers connected to departing employees.
The authors use a real dataset of internal communications from a major company covering communications among 100K employees.
Their empirical study demonstrates the effects of a departure in the socialization of peers.
They provide insights into managing workforce changes and preserving communication dynamics in the face of employee exits.







(+) The motivation of the paper is clear.

(+) The paper targets an important problem.

(+) The authors conduct experiments on real data.


(-) The experiment seems reasonable, although the dataset is not available for reproduction.

(-) The experiments are somewhat limited.

(-) The technical contribution of the work is somewhat unclear.




The paper addresses an important issue.
It provides a good brief survey of the related work.
The experiment seems reasonable, although the dataset is not available for reproduction.


(additional comments)

The technical contribution of the paper is relatively unclear.
For example, the framework developed seems based on graph structural analysis.
The related research study has attracted much attention recently, and many approaches have been developed.
I would like to see more discussion about the differences and novelty compared to other approaches, including time-evolving graph analysis.

There are some [?] in the text, e.g., lines 166, 839, 1244.

**Questions:**

N/A

**Reviewer Confidence:**

2: The reviewer is willing to defend the evaluation, but it is likely that the reviewer did not understand parts of the paper

**Scope:**

3: The work is somewhat relevant to the Web and to the track, and is of narrow interest to a sub-community

---

### Decision · Program_Chairs · 2024-01-22

**Decision:**

Accept

**Comment:**

Summary: Investigates the impacts of employee exits on socialization patterns among remaining coworkers.

 Strengths:
 + Well motivated an addressing an important but less studied problem
 + Experiments on real communication data
 + Large-scale study with extensive experiments
 + Some insightful comparison across stress periods
 + Well-structured and well-written

 Weaknesses:
 - Lack of availability of the dataset for reproduction
 - Limited scope in experiments
 - Technical contribution of the work is somewhat unclear
 - Results might benefit from better statistical grounding
 - Limitation in generalizability beyond single company

 Recommendation: While a first-of-its kind study, the submission was found to be limited in generalizability and technical contributions.